# A Tutorial on How to Conduct Meta-Analysis with IBM SPSS Statistics

Sedat Sen [1,*] and Ibrahim Yildirim [2,*]

1    Education Faculty, Harran University, Sanliurfa 63190, Turkey
2    Education Faculty, Gaziantep University, Gaziantep 27310, Turkey
*    Correspondence: sedatsen06@gmail.com (S.S.); iyildirim84@gmail.com (I.Y.)

**Abstract:** Meta-analysis has started to take place among the most used methodologies in psychological research. Such a technique allows researchers to combine the data sets obtained from several individual studies on the same topic and thus is particularly useful for finding solutions to controversial issues that cannot be solved with individual studies. This paper presents a detailed tutorial of the IBM SPSS software, which enables one to implement the statistical analyses for meta-analysis. Examples are also provided to highlight the main analyses conducted in the meta-analysis. The tutorial ends by discussing the differences between IBM SPSS capabilities and those of other software packages.

**Keywords:** meta-analysis; meta-regression; IBM SPSS; software

## 1. Introduction

Scientific research is a cumulative process in which each scientist makes unique contributions to their area of study. After a certain period of time, these individual studies may reveal different findings about the subject studied. When we look at the research on a particular subject as a whole, we may not be able to see whether the methods applied or developed are really effective. An example of this situation was experienced in the field of psychotherapy in the 1950s. In 1952, Hans Eysenck initiated a fierce debate in clinical psychology by publishing a study arguing that psychotherapy had no beneficial effect on patients [1]. By the mid-1970s, hundreds of psychotherapy studies had produced a dizzying array of positive, neutral, and negative results, and reviews of these studies failed to settle the debate. To evaluate Eysenck's claim, Gene V. Glass calculated an overall mean value for 375 psychotherapy studies by statistically standardizing the differences between treatment and control groups. Smith and Glass [2] published their findings in a journal and showed that psychotherapy was actually an effective practice. Glass called this method "meta-analysis". Despite criticism from some scientists [3], meta-analysis is now accepted as an appropriate method of statistically summarizing the results of individual quantitative studies in the behavioral, social, and health sciences [4]. Although the term meta-analysis was first used by Glass in 1976, the first meta-analysis in the sense of combining quantitative studies is attributed to Pearson [5], who analyzed data from five studies on the correlations between inoculation and immunity and mortality. In the late 1970s and early 1980s, following Glass's work, among others, Rosenthal [6], Glass, McGaw and Smith [7], Hedges [8,9], Hunter, Schmidt, and Jackson [10], and Light and Pillemar [11] popularized meta-analysis and further developed the statistical methods necessary for its application.

It is known as systematic review, in which scientists systematically review the results from a large number of studies and synthesize the results in order to make inferences about the typical findings and sources of variability between studies. Over the past 40 years, there has been a large increase in the use of systematic reviews in both medicine and the social sciences, including psychology and education. The focus on evidence-based practice

in many professions has increased interest in understanding both the known and unknown parts of important interventions and clinical practice [12]. Systematic reviews promise a transparent and repeatable method for summarizing the literature to help improve both policy decisions and the design of new studies. Although systematic reviews have a certain potential, this potential is also observed to be compromised by inadequate methods and misinterpretation of results [12]. In short, a systematic review is a critical evaluation to seek the answer to a focused question in the light of available research. However, meta-analysis differs from systematic review in that it only focuses on quantitative studies. The present study focuses on the meta-analysis method, which was developed based on quantitative research and has emerged as a methodological and statistical approach to draw conclusions from the empirical literature.

Meta-analysis is a quantitative method used to combine the results of multiple studies into a single conclusion. The term "meta-analysis" was first coined by Gene Glass in 1976 as the statistical analysis of a large collection of analysis results from individual studies for the purpose of integrating the findings [13] (p. 3). A meta-analysis collects quantitative results from multiple studies and draws conclusions about the overall effect between studies. In doing so, it does not look at what results the original studies found. The word "meta" is used because it is a kind of research of research or analysis of analysis [13]. To put it briefly, it is a systematic quantitative research method to reveal the big picture of a topic. In order to conduct a meta-analysis study, Glass suggested using the effect size value when combining the findings of multiple studies correctly [13,14]. Any standardized index (standardized mean difference, correlation, and odds ratio) can be used as an effect size as long as it is comparable between studies, independent of the sample, and indicates the size and direction of the effect. The effect size is the value that makes meta-analysis possible. The effect size is taken as the "dependent variable" in the meta-analysis and results in a comparable statistic as it is obtained by standardizing between studies.

The general purpose of meta-analysis is to combine the results of individual studies to reach summary conclusions about a research question. It is used to calculate a summary estimate of effect size, to explore the causes of differences in effects between studies, and to identify heterogeneity in the effects (or differences in risk) of the intervention in different subgroups. It is worth mentioning here that the meta-analysis calculates the weighted average of the effect size, not the arithmetic mean between studies. It is an approach that gives more weight to more precise estimates. In other words, it gives greater weight to studies with a large sample size. The weighting factor is equal to $1/(\text{standard error})^2$. Studies with a low standard error (i.e., large sample size) contribute more to the overall average estimated as a result of the meta-analysis.

In meta-analysis, the overall average estimate can be typically obtained with either a fixed-effect or random-effects model. The model assuming that the parameter measuring the effect size is the same in all studies is called the "fixed-effect" model. The model that allows this parameter to act as a random variable that takes different values from one study to another is called the "random-effects" model. The fixed effect model and the random-effects model make different assumptions and apply different weights in the calculation of the average effect size. There is only one source of variation (i.e., the sampling error) in the fixed-effect model. That is, the difference between each effect size is due to the difference in sample size, and the population effect size is the same for each study. It is assumed that each effect size value in the study comes from a fixed population. On the other hand, there are two sources of variation in the random-effects model. The random-effects model assumes that each observed effect size differs from the population mean by an individual-level sampling error plus a value representing other sources of variability assumed to be randomly distributed. Although there are different ways of performing meta-analysis, the most common and popular approaches are those offered by Hunter and Schmidt [10,15,16], Glass [7,13], and Hedges and Olkin [17]. All three approaches aim to transform the results of individual studies into a common measure.

An excellent literature review is at the heart of the meta-analysis. A common threat to literature reviews and meta-analyses is known as publication bias. The term "publication bias" is often used to express that statistically significant results are more likely to be presented and published than non-significant and null results [18]. Publication bias is a systematic error that occurs in a statistical inference conditioned on gaining publication status [19]. As a matter of fact, published studies alone do not represent all studies in a research area. This situation is also called the file drawer problem [20]. It is seen as a threat because it adds systematic error to the meta-analysis. This threat arises because studies that have not found a statistically significant effect (or have not found the expected effect) are less likely to be published and therefore less likely to be available to the meta-analyst. Lipsey and Wilson [21] provided evidence for publication bias by showing that published studies had a larger mean effect size than unpublished studies. A study group included in a meta-analysis may be over-representative of published studies, as it is much easier to identify and screen published studies than unpublished studies that were never written due to negative or null findings. Another source of bias is the presence of gray literature [22]. This includes conference presentations, technical reports, or obscure publications, and is kept between the drawer and the publication process. This situation is also referred to as "fugitive literature" by Rosenthal [23]. In order to eliminate publication bias, a comprehensive search should be conducted to find these missing studies. To counter this threat, one should seek to obtain unpublished work (for example, dissertations and conference proceedings) that will either eliminate this threat or at least allow one to assess the magnitude of this bias. That is, the primary way to avoid publication bias in meta-analysis is to include both published and unpublished studies. Card (2011) states six methods that can be used to examine whether there is a publication bias. These are moderator analysis, funnel plot, fail-safe N, regression analysis, trim and fill, and weighted selection methods. Apart from these, there is another method proposed by Begg and Mazumdar [24] based on rank correlations.

### 1.1. Steps of Meta-Analysis

Researchers who want to perform these analyses through meta-analysis should follow certain steps. Although it is presented in different ways in many sources, the steps required to perform a meta-analysis can be listed as follows:

- The research question should be formulated.
- A decision should be made on how to select appropriate studies from the collected studies.
- Appropriate studies should be collected according to research questions and keywords.
- Quality control/sensitivity analyses should be done.
- The effect size to be used in the selected studies should be decided and calculated for each study.
- The data should be pooled and a summary measure and confidence interval should be calculated.
- Additional analyses (heterogeneity, publication bias, etc.) should be done.
- Moderator analyses for moderator variables should be performed.
- Results should be interpreted and inferences should be made.
- In addition, the details of the above-mentioned steps should be reported together with the meta-analysis findings.

Meta-analysis is used extensively in education, psychology, health, and several other areas to summarize the results of individual studies conducted on the same topic. This method helps researchers to estimate the mean effect size using the effect size and variance (or standard error) values from each individual study. While this method may seem straightforward, statistical analyses of meta-analysis data resulting from individual studies often present great challenges. Thus, several software packages have been developed for this purpose.

### 1.2. Software Options

Meta-analysis studies are very demanding in terms of both data collection and data analysis. It is a very time-consuming process to identify the studies that will be included in the meta-analysis and to extract the necessary information for calculating the effect size. In light of the summary information obtained, calculating the effect size value for each study and obtaining the overall mean is another time-consuming step where researchers with poor statistical knowledge are likely to make mistakes. Unfortunately, while the creation of the data file specified here is mandatory by the researcher himself, there are several software packages developed to perform the second step, the data analysis process. Researchers have two options when it comes to software that can be used—using specialized software designed for meta-analysis (e.g., Comprehensive Meta-Analysis) or using statistical software designed for general purposes (e.g., SPSS).

In the literature, several standalone software packages have been made available, especially for meta-analysis. Commercial packages include MetaWin [25] and Comprehensive Meta-Analysis (CMA) [26]. DSTAT [27] and Advanced Basic Meta-analysis [28] are other commercial software programs that are less well-known than MetaWin and CMA. In addition to these software packages, free meta-analysis specific software packages are also available, including RevMan (Review Manager), MetaGenyo [29], MetaStat [30], Meta-Analysis [31], META (Meta-Analysis Easy to Answer) [32], and OpenMeta [Analyst] [33].

Some of the existing software packages have also been expanded for meta-analysis. Examples of these packages are MIX 2.0 [34], metaXL [35], and MetaEasy [36] add-ins developed for Excel. Functions and macros have also been prepared for meta-analysis in Stata [37]. Using the proc mixed command, meta-analysis can be conducted through the SAS program [38]. Various meta-analysis packages are also available in R [39] including meta [40], metafor [41], rmeta [42], robumeta [43] and metaSEM [44]. Detailed information about other meta-analysis packages in the R program can be found in the study of Polanin, Hennessy and Tanner-Smith [45]. A module called MAJOR in Jamovi, developed by Kyle Hamilton, allows users to conduct a meta-analysis using different types of input (e.g., effect sizes, correlation coefficients). Similarly, another open-source statistical software called JASP can also be used for meta-analysis. The engine behind these two software packages comes from the R package metafor. In addition, some macros have been developed to conduct meta-analysis using the SPSS program [4,46]. The existing SPSS macros, however, currently only provide limited capabilities for conducting analyses and enable researchers to conduct only main analyses (i.e., mean ES, subgroup analyses, and meta-regression analyses). Publication bias and other graphical options (e.g., forest plot and funnel plot) were not available in these SPSS macros (see also [47]). SPSS macros also require researchers to write SPSS syntax, which would be cumbersome for most practitioners. Very recently, IBM SPSS introduced a point-and-click meta-analysis menu with Version 28. Although many programs have been developed in the literature, the SPSS program remains the first choice for many researchers. Thus, researchers familiar with using SPSS may want to conduct the statistical analyses required for meta-analysis via SPSS. To date, no study has been conducted on how to conduct meta-analysis with IBM SPSS. The tutorial in this study provides guidance for students and researchers who originally plan to use IBM SPSS for meta-analysis of the data collected from individual studies.

### 1.3. Properties of IBM SPSS Statistics

It is possible to conduct most of the analyses required for meta-analysis studies using IBM SPSS Statistics with Version 28 (SPSS28). The trial version of SPSS28 can be downloaded from the official website (https://www.ibm.com/products/spss-statistics) (accessed on 31 July 2022). After clicking the "Try SPSS Statistics at no cost" link, to start the trial period, you should enter some information (e.g., name, e-mail address). With this information, you can obtain an IBMid and code. With this code you can set up SPSS28 on your PC. The trial period is limited to 30 days. After the trial period, one may want to purchase the software.

Whether you have the demo or the full version, SPSS28 has several procedures, including mean effect size calculation, heterogeneity statistics, publication bias, and moderator analyses.

### 1.3.1. Main Analyses

There are three main submenus under the Meta Analysis menu of SPSS28: Continuous Outcomes, Binary Outcomes, and Meta Regression. Users can perform meta-analysis with either continuous or binary outcomes on raw data. In addition, similar analyses can be performed when the pre-calculated effect size data are available with continuous or binary outcomes. These are presented in the Continuous Outcomes and Binary Outcomes submenus. The users with summary data (e.g., N, mean, and SD) should use the Raw Data submenu, and the users with pre-calculated effect sizes (ES and its variance) should use the Pre-calculated Effect Size submenu. Both fixed-effect and random-effects models are available under the model section. Users can also conduct subgroup analyses under these menus.

When the Raw Data submenu of the Continuous Outcomes menu is selected, the Effect Size section has four types of effect size indices: Unstandardized mean difference, Cohen's *d*, Hedges' *g*, and Glass' delta. In addition to Study ID, the summary statistics required for meta-analyses are sample size, mean, and standard deviation (or variance) values for both control and treatment groups. When the Raw Data submenu of the Binary Outcomes menu is selected, the Effect Size section has four types of effect size indices: Log Odds Ratio, Peto's Log Odds Ratio, Log Risk Ratio, and Risk Difference. In addition to Study ID, the summary statistics required for meta-analyses are success and failure rates for both control and treatment groups. When the Pre-Calculated Effect Size submenu of the Continuous Outcomes or Binary Outcomes menus is selected, the Effect Size and its standard error or variance should be selected. In addition, one of the effect size types (Log Odds Ratio, Peto's Log Odds Ratio, Log Risk Ratio, and Risk Difference) should be selected for binary outcomes data.

Whichever of the data entry types aforementioned above you choose, you can modify several options, including Criteria, Analysis, Inference, Contrast, Bias, Trim-and-Fill, Print, Save, and Plot menus (see Figure 1). The Criteria dialog has several options for confidence interval, missing data, iteration, and convergence. Cumulative analysis and subgroup analysis can be selected under the Analysis submenu. Estimator type and standard error adjustment can be determined under the Inference dialog. Currently, there are seven estimators available in SPSS: Restricted maximum likelihood (REML), which is the default, Maximum likelihood (ML), Empirical Bayes, Hedges, Hunter–Schmidt, DerSimonian–Laird, and Sidik–Jonkman. The standard error adjustment dialog includes three options: no adjustment, Apply the Knapp–Hartung adjustment, and Apply the truncated Knapp–Hartung adjustment. As mentioned on the IBM website, the Contrast dialog provides settings for controlling the contrast test for meta-analysis with continuous outcomes on raw data that are provided in the active data set for the estimation of the effect size.

### 1.3.2. Publication Bias Analyses

Several analyses for publication bias assessment can be applied in SPSS28. Egger's regression test can be applied via Bias dialog, while the trim-and-fill method can be performed with the Trim-and-Fill dialog. A funnel plot can also be obtained in SPSS28 to examine whether the relationship between standard errors and effect sizes shows a symmetrical shape. In addition to Funnel Plot, several plots, including Forest Plot, Cumulative Forest Plot, Bubble Plot, and Galbraith Plot, can be created via this Plot dialog. The Print dialog can be used to show test of homogeneity and heterogeneity statistics in the output screen. The Print dialog also enables users to print effect size and prediction intervals. Lastly, the Save dialog can be used to save several statistics including individual effect size, standard error, confidence interval lower bound, confidence interval upper bound, *p*-value, study weight, and percentage of study weight (Please visit IBM website for more details:

https://www.ibm.com/docs/en/spss-statistics/SaaS?topic=features-meta-analysis) (accessed on 30 July 2022).

**Figure 1.** Screenshot of the Continuous Outcomes Submenu.

### 1.3.3. Subgroup Analyses and Meta-Regression

SPSS28 enables users to conduct both subgroup analyses and meta-regression. Subgroup analysis can be selected under the Analysis submenu. In order to conduct a subgroup analysis, users need to add the variables of interest (categorical moderator) from the 'Variables' box into the 'Subgroup Analysis' box. Using the Meta Regression submenu, users can perform meta-regression analyses by selecting the effect size, standard error (or variance and weight), factor(s), and covariate(s). Categorical moderators are listed in factor(s) and numeric variables are listed in covariate(s). The Meta Regression submenu has some of the dialogs mentioned above: Criteria, Inference, Print, Save, and Plot. While options under the Criteria and Inference dialogs remain the same, the Print dialog allows users to display exponentiated statistics and model coefficient tests. Save dialog enables users to save several statistics, including predicted values, standard error of predicted values, confidence interval lower bound, confidence interval upper bound, residuals, standard error of residuals, leverages, fixed linear predictions, standard error of fixed linear predictions, best linear unbiased predictions (BLUPs), and standard error of BLUPs. A Bubble plot can also be created using the Plot dialog under the Meta Regression submenu.

### 1.4. Steps of Conducting Meta-Analysis in IBM SPSS Statistics

Step 1: Prepare your data set

In order to conduct a meta-analysis, the effect size and variance (or standard error) of each individual study should be collected or calculated. Sometimes, researchers may have only summary data instead. SPSS28 allows users to save data as pre-calculated effect size or summary data. The researchers planning to perform a meta-analysis based on continuous data should collect sample size ($N$), mean, and SD values for both the control and treatment groups of each study. The researchers planning to perform a meta-analysis based on binary data should collect success and failure rates for both the control and treatment groups

of each study. However, the researchers planning to perform a meta-analysis based on correlation should collect correlation and sample size for each individual study. Then, Pearson correlation coefficients should be transformed to Fisher's *z* values. In addition, the variance or standard error of Fisher's *z* values should be computed. For continuous and binary data, pre-calculated effect size and its variance or standard error can also be calculated and saved in a data set. Table 1 shows an example data file for pre-calculated effect size for correlation data. The example data file for raw data (means, SDs, and *N*s) is demonstrated in Table 2. Each variable measured is represented by a column, and each measurement of that variable is represented by a row in what is known as a wide data format. A variable representing study ID would usually be placed in the first column, followed by the variables defining the effect size and its variance, or summary variables. Study in tables are explained in Supplementary Materials.

**Table 1.** Sample Dataset for Correlation and Moderator Analyses [48].

| Study | *n* | *r* | *vr* | Country | Region | Business | Female % | *z* | *vz* |
|---|---|---|---|---|---|---|---|---|---|
| Chen & Yang, 2012 | 227 | 0.365 | 0.003 | Taiwan | FE | Others | 49.4 | 0.383 | 0.004 |
| Kaya, 2015 | 383 | 0.600 | 0.001 | Turkey | ME | Education | 50.5 | 0.693 | 0.003 |
| Hunsaker, 2016 | 263 | 0.500 | 0.002 | South Korea | FE | Others | 33 | 0.549 | 0.004 |
| Wu & Li, 2015 | 239 | 0.470 | 0.003 | Taiwan | FE | Others | 38.6 | 0.510 | 0.004 |
| Bozkurt & Töremen, 2015 | 409 | 0.524 | 0.001 | Turkey | ME | Education | 52.81 | 0.582 | 0.002 |
| Nafei, 2018 | 285 | 0.603 | 0.001 | Egypt | ME | Others | 60 | 0.698 | 0.004 |
| Çimen, 2016 | 301 | 0.430 | 0.002 | Turkey | ME | Education | 56.48 | 0.460 | 0.003 |
| Göçen & Kaya, 2020 | 102 | 0.490 | 0.006 | Turkey | ME | Education | 22.56 | 0.536 | 0.010 |
| Chen & Li, 2013 | 122 | 0.433 | 0.006 | China | FE | Others | 0.8 | 0.464 | 0.008 |
| Phuong et al., 2018 | 329 | 0.285 | 0.003 | Vietnam | FE | Others | 45 | 0.293 | 0.003 |

Note. *n* = Sample size, *r* = Pearson correlation, *vr* = variance of Pearson *r*, *z* = Fisher *z*, *vz* = variance of Fisher *z*.

**Table 2.** Sample Raw Dataset [49] for Standardized Mean Difference Example.

| Study | Blended *n* | Blended Mean | Blended SD | Face-to-Face *n* | Face-to-Face Mean | Face-to-Face SD |
|---|---|---|---|---|---|---|
| Unsal, 2007 | 24 | 31 | 2.5 | 22 | 31.05 | 2.82 |
| Turkcapar, 2011 | 28 | 18.14 | 3.67 | 28 | 15.89 | 4.67 |
| Aygun, 2011 | 35 | 18.91 | 2.72 | 36 | 15.23 | 4.003 |
| Aksogan, 2011 | 32 | 53.8 | 11.9 | 31 | 50.25 | 16.76 |
| Yapici, 2011 | 47 | 25.11 | 5.04 | 60 | 19.08 | 2.657 |
| Yildiz, 2011 | 36 | 8.41 | 0.996 | 35 | 7.6 | 1.03 |
| Turk, 2012 | 51 | 71.57 | 13.47 | 64 | 58.36 | 14.28 |
| Saritepeci, 2012 | 52 | 12.36 | 4.11 | 55 | 10.25 | 4.1 |
| Demirkol, 2012 | 27 | 78.7 | 13.05 | 27 | 72.22 | 9.12 |
| Akgündüz, 2013a | 25 | 20.44 | 5.874 | 24 | 15.792 | 6.29 |
| Akgündüz, 2013b | 25 | 18.08 | 6.211 | 24 | 15.792 | 6.29 |
| Pesen, 2014a | 38 | 32.23 | 2.87 | 38 | 29.86 | 2.56 |
| Pesen, 2014b | 41 | 28.17 | 3.77 | 41 | 28.43 | 3.16 |
| Kahyaoglu, 2014 | 25 | 31.44 | 3.78 | 25 | 26 | 8.14 |

Step 2: Open IBM SPSS Statistics and import your data set

Researchers either prepare data sets in the SPSS program or save them in other file formats such as MS Excel. In the case of other data formats, the data file should be imported into the SPSS program. For example, to import the meta-analysis data from Excel to SPSS:

Select File > Import Data > Excel.

It is important to click on the "read variable names from first row of data" box when you have the variable names on the first row of Excel file. As an alternative, users can enter the data on the blank page opened in the variable and data view sections in the SPSS program.

Step 3: Open Meta Analysis menu

The Meta Analysis procedure performs meta-analysis on the data in the active data set in order to estimate the overall effect size. To open the Meta Analysis menu (see Figure 2): Select Analyze > Meta Analysis.

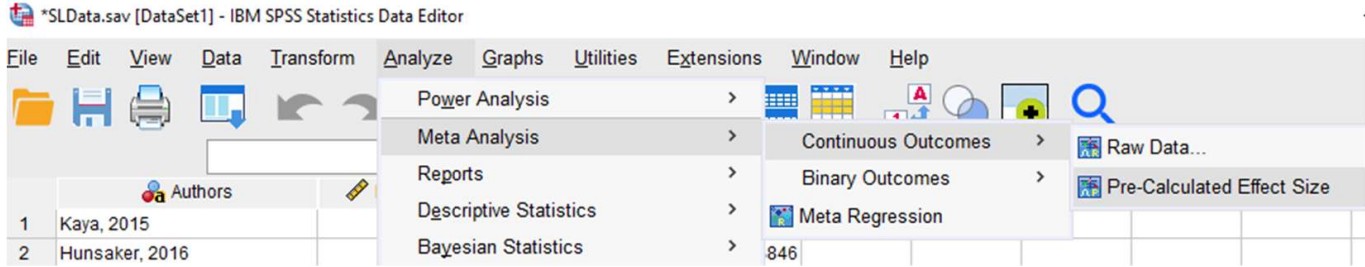

**Figure 2.** SPSS Meta Analysis Menu.

Step 4: Calculate mean effect size

There are several options under the Meta Analysis menu. As a result, the researchers should choose the option that best fits their data set and research question. The choice depends on the data type (continuous or binary) and whether it is summary or pre-calculated effect size. To be able to analyze a continuous raw data set:

- Select Analyze > Meta Analysis > Continuous Outcomes > Raw Data . . .
- Add the variables of the treatment group (sample size, mean, and SD) into the 'Treatment Group' box
- Add the variables of the control group (sample size, mean, and SD) into the 'Control Group' box
- Add the identifying variable (Authors' names) into the 'Study ID' box
- Select the effect size type (Cohen's *d*, Hedges' *g*, Glass' delta, Unstandardized Mean Difference) in the 'Effect Size' box
- Select the model type (fixed-effect or random-effects) under the 'Model' box
- Click 'OK'

Similar steps can be followed for binary data sets. Users need to add success and failure variables into the treatment group and control group boxes. The effect size type would be one of the following: Logg Odds Ratio, Peto's Logg Odds Ratio, Logg Risk Ratio, and Risk Difference.

To be able to calculate the mean effect size with pre-calculated effect sizes:

- Select Analyze > Meta Analysis > Binary Outcomes > Pre-Calculated Effect Size
- Select the effect size type (Logg Odds Ratio, Peto's Logg Odds Ratio, Logg Risk Ratio, and Risk Difference) in the 'Effect Size' box
- Add the effect size variable (e.g., Logg Odds Ratio) into the 'Effect Size' box
- Add the variance variable into the 'Variance' box
- Alternatively, add the standard error variable into the 'Standard Error' box
- Add the identifying variable (Authors' names) into the 'Study ID' box
- Select the model type (fixed-effect or random-effects) under the 'Model' box
- Click 'OK'

When these steps are applied, the mean effect size and other statistics will be shown in three tables (Meta-Analysis Summary, Case Processing Summary, and Effect Size Estimates) as a part of the output. The table labeled as 'Effect Size Estimates' shows the mean effect size, its standard error, Z-value, two-tailed *p*-value, and 95% confidence interval.

Step 5: Check heterogeneity

One of the important analyses requires an assessment of heterogeneity. This can be checked with several statistics, including Q-statistics, Tau-squared, H-squared, and I-squared in SPSS28. To be able to obtain heterogeneity statistics:

- Select Analyze > Meta Analysis > Continuous Outcomes > Raw Data . . .
- Click on the 'Print' dialog
- Check the dialog boxes labeled as 'Test of homogeneity' and 'Heterogeneity measures'
- Click 'Continue' to go back to the main screen
- Click 'OK'

When these steps are applied, the heterogeneity statistics in two tables (Test of Residual Homogeneity and Residual Heterogeneity) will be shown in the output.

Step 6: Create plots

Another way of checking the heterogeneity is to create some plots. For example, a forest plot can be examined to do a visual assessment of heterogeneity. To be able to create a forest plot:

- Select one of the input screens under the Meta Analysis menu
- For example, Select Analyze > Meta Analysis > Continuous Outcomes > Raw Data
- Click on the 'Plot' dialog
- Select 'Forest Plot'
- Check the dialog box under the 'Display Columns'
- Decide 'Position of plot column', 'Annotations', and 'Reference lines'
- Click 'Continue' to go back to the main screen
- Click 'OK'

When these steps are applied, the forest plot will be shown in the output. Other plots, including cumulative forest plot, bubble plot, funnel plot, and Galbraith plot, can be obtained using the 'Plot' dialog under the Meta Analysis menu.

Step 7: Assess publication bias

A meta-analyst should ensure that publication bias is not an issue for the studies included in the meta-analysis. This can be examined using several statistics in SPSS28, including funnel plot, Egger's regression test, and trim-and-fill methods. The funnel plot can be obtained using the 'Plot' dialog described in the previous step. To be able to perform Egger's regression test:

- Select one of the input screens under the Meta Analysis menu
- For example, Select Analyze > Meta Analysis > Continuous Outcomes > Raw Data
- Click on the 'Bias' dialog
- Select 'Egger's regression-based test'
- Check the dialog boxes under the 'Include intercept in regression' and 'Estimates statistics based on t-distribution'
- Click 'Continue' to go back to the main screen
- Click 'OK'

When these steps are applied, the results of the Egger's regression test will be shown in the output. To be able to perform trim-and-fill method:

- Select one of the input screens under the Meta Analysis menu
- For example, Select Analyze > Meta Analysis > Continuous Outcomes > Raw Data
- Click on the 'Trim-and-Fill' dialog
- Select 'Estimate number of missing studies'
- Check the dialog boxes under the 'Side to Impute Studies' as left or right. Another option is to click on 'Determined by the slopes of Egger's test'
- Other options can be determined by clicking the boxes under 'Method', 'Iteration Process'
- Click 'Continue' to go back to the main screen
- Click 'OK'

When these steps are applied, the results of the trim-and-fill method will be shown in the output.

Step 8: Perform subgroup analyses

A meta-analyst should examine the possible source of heterogeneity in the case of lack of homogeneity among the individual studies. To do this, subgroup analyses can be applied using the categorical moderators (e.g., publication type) collected from individual studies. To be able to perform subgroup analysis:

- Select one of the input screens under the Meta Analysis menu
- For example, Select Analyze > Meta Analysis > Continuous Outcomes > Raw Data
- Click on the 'Analysis' dialog
- Add the variables of interest (categorical moderator) from the 'Variables' box into the 'Subgroup Analysis' box
- Click 'Continue' to go back to the main screen
- Click 'OK'

When these steps are applied, the results of the subgroups of the categorical variable will be shown in the output. For each category, the table labeled as 'Effect Size Estimates for Subgroup Analysis' will show the mean effect size, its standard error, Z-value, two-tailed *p*-value, and 95% confidence interval and prediction interval. Overall results will be reported in the last row of the table.

Step 9: Perform meta-regression analyses

Another way of examining the possible source of heterogeneity is to conduct a meta-regression analysis using continuous (e.g., mean age of the sample) and categorical moderators (e.g., publication type) collected from individual studies. However, subgroup analysis can be done with only categorical variables. With meta-regression analysis, researchers can analyze both continuous and categorical moderators. This method also allows us to include more than one moderator in the regression model. To be able to perform meta-regression analysis:

- Select Analyze > Meta Analysis > Meta Regression
- Add the effect size variable (e.g., Cohen's *d*) from the 'Variables' box into the 'Effect size' box
- Add the effect size variance from the 'Variables' box into the 'Variance' box. Alternatively, one can use the standard error or weight of the effect size
- Add the variables of interest (continuous moderator) from the 'Variables' box into the 'Covariate(s)' box
- Add the variables of interest (categorical moderator) from the 'Variables' box into the 'Factor(s)' box
- Click 'Continue' to go back to the main screen
- Click 'OK'

When these steps are applied, the results of the meta-regression analysis will be shown in the tables (Model Summary, Case Processing Summary, Model Coefficient Test, and Parameter Estimates) as a part of the output. The table labeled as 'Parameter Estimates' shows the regression coefficient, its standard error, t-value, two-tailed *p*-value, and 95% confidence interval. As mentioned above, the Meta Regression submenu has several dialogs, including Criteria, Inference, Print, Save, and Plot. These dialogs can be used to obtain additional information such as bubble plots, diagnostic statistics, etc.

## 2. Empirical Examples

### 2.1. Example 1 (Standardized Mean Difference)

In this section, we will present an example of applying a standardized mean difference-based meta-analysis containing two group comparisons. For this purpose, we used the sample data retrieved from Çırak Kurt, Yıldırım, and Cücük's [49] study (see Table 2). The sample data set includes student achievement comparisons in blended learning and

face-to-face learning environments. Additionally, the data set includes only 14 studies and post-test scores of students. Çırak Kurt et al. [49] collected sample size, mean, and standard deviation values for experimental (blended learning) and control (face-to-face learning) groups.

With the values presented in Table 2, the Cohen's *d* value (standardized mean difference) can be calculated for each study as follows:

$$\text{Cohen's } d = \frac{\bar{x}_{exp.} - \bar{x}_{control}}{\text{SD}_{pooled}} \tag{1}$$

Cohen's *d* value is calculated by dividing the difference between means by the pooled standard deviation (*SD*$_{pooled}$) that can be calculated as below:

$$SD_{pooled} = \sqrt{\frac{(n_{G1} - 1)s_{G1}^2 + (n_{G2} - 1)s_{G2}^2}{(n_{G1} - 1) + (n_{G2} - 1)}} \tag{2}$$

where $n_{G1}$ and $n_{G2}$ are sample sizes of control and treatment groups and $s_{G1}^2$ and $s_{G2}^2$ are variances. For Cohen's *d*, the standard error can be calculated as below:

$$\text{S.E.} = \sqrt{\frac{n_{G1} + n_{G2}}{n_{G1}n_{G1}} + \frac{(d)^2}{2(n_{G1} + n_{G2})}}. \tag{3}$$

Another standardized mean difference index is named Hedges' *g*, which applies a correction for bias due to small sample sizes as follows:

$$\text{Hedges' } g = \text{Cohen's } d \times \left(1 - \frac{3}{4(n_{exp.} + n_{control}) - 9}\right) \tag{4}$$

Although Glass' delta is a less preferred effect size, it is used in some studies. Glass' delta assumes that the standard deviations are different between groups. Additionally, Glass' delta only uses the standard deviation of the control group [50]. Glass' delta and its variance can be calculated as follows:

$$\text{Glass' delta} = \frac{\bar{x}_{exp.} - \bar{x}_{control}}{\text{SD}_{control}} \tag{5}$$

$$v_{\text{Glass' delta}} = \frac{n_{exp} + n_{control}}{n_{exp}.n_{control}} + \frac{\text{Glass' delta}^2}{2(n_{control} - 1)} \tag{6}$$

The sample analyses here were conducted with Hedges' *g* value. However, similar analyses can be applied with Cohen's *d* and Glass' delta. To obtain a mean Hedges' *g* value, there are two data entry options in SPSS: raw data or pre-calculated effect sizes. If one calculates effect size values from online platforms (e.g., https://www.campbellcollaboration.org/research-resources/effect-size-calculator.html) (accessed on 30 July 2022), the following steps can be used:

- Select Analyze > Meta Analysis > Continuous Outcomes > Pre-Calculated Effect Size

An easier way to do this is to conduct the analysis using raw data when you have the data ready as entered in Excel. For this option, the following steps can be used (see Figure 3):

- Select File > Open > Data
- Select "Files of Type" as "Excel"
- Find the data > Open
- Select Analyze > Meta Analysis > Continuous Outcomes > Raw Data (see Figure 4)

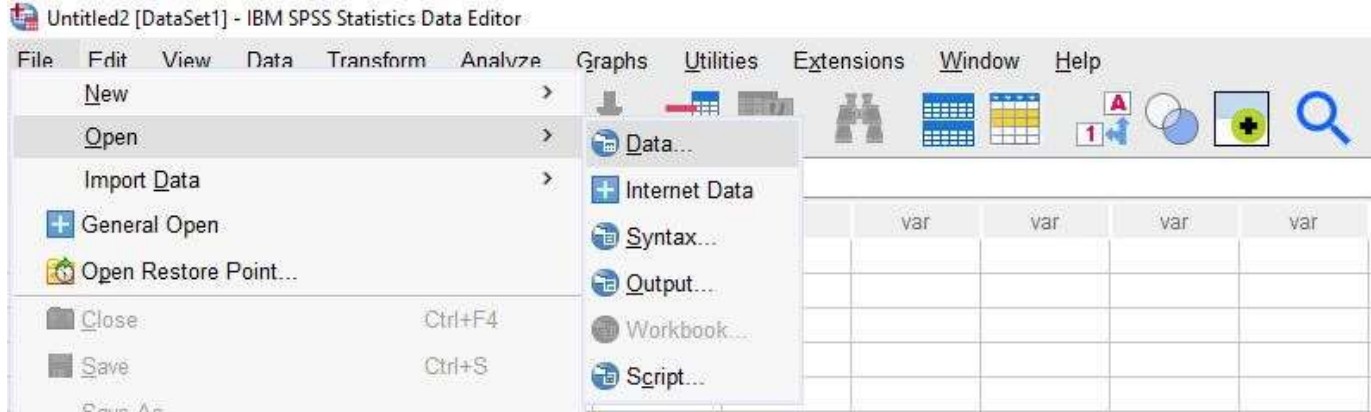

**Figure 3.** Open Data Menu.

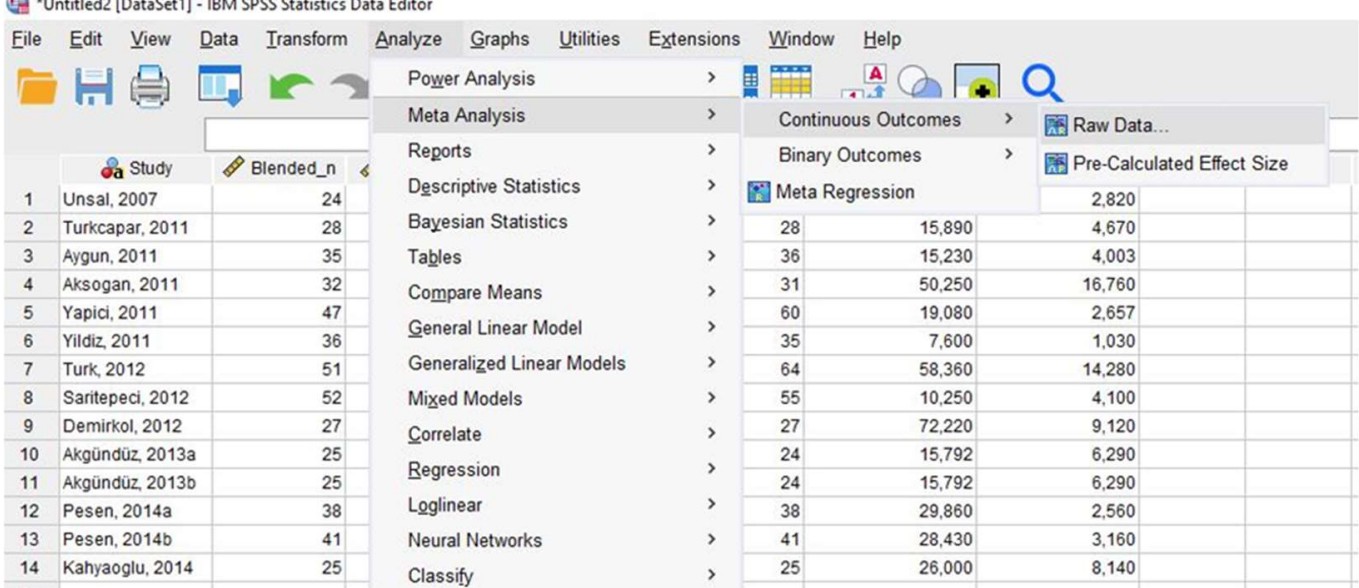

**Figure 4.** Meta-analysis with Raw Data.

- Add the variables of the treatment group (sample size, mean, and SD) into the 'Treatment Group' box
- Add the variables of the control group (sample size, mean, and SD) into the 'Control Group' box
- Add the identifying variable (Authors' names [Study]) into the 'Study ID' box
- Select the effect size type as 'Hedges' *g*' in the 'Effect Size' box
- Select the model type 'Random-effects' under the 'Model' box
- Open 'Print' Dialogue > Select the 'Test of homogeneity' and 'Heterogeneity Measures' > Click 'Continue' (see Figure 5)
- Open 'Plot' Dialogue > Select 'Forest Plot' box and all the 'Display Columns' boxes > Click 'Continue'
- Click 'OK' (see Figure 6)

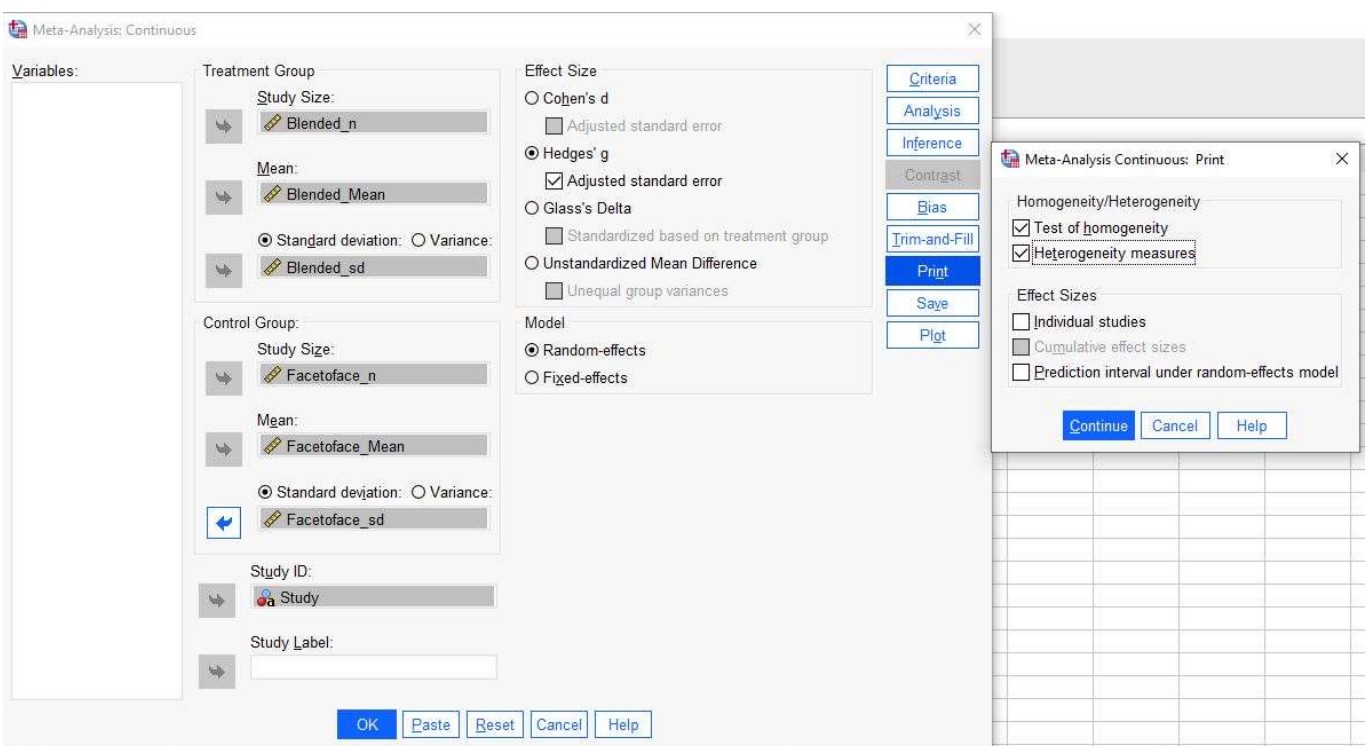

**Figure 5.** Data Identification Menu.

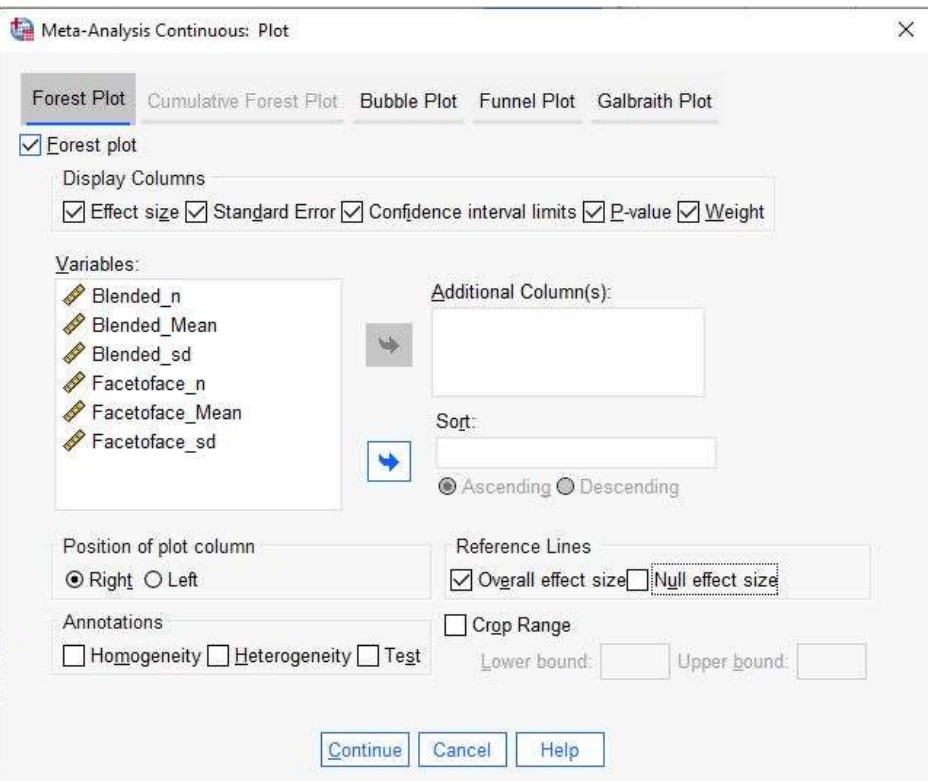

**Figure 6.** Forrest Plot Menu.

When these steps are applied, the outputs will be presented in the new window as in Figure 7.

**Meta-Analysis Summary**

| Data Type | Raw |
| --- | --- |
| Outcome Type | Continuous |
| Effect Size Measure | Hedges' g |
| Model | Random-effects |
| Weight | Inverse-variance[a] |
| Estimation Method | REML |
| Standard Error Adjustment | None |

a. Random-effects weights including both within-
and between-study variance.

**Effect Size Estimates**

| | Effect Size | Std. Error | Z | Sig. (2-tailed) | 95% Confidence Interval Lower | Upper |
| --- | --- | --- | --- | --- | --- | --- |
| Overall | ,644 | ,1190 | 5,417 | <,001 | ,411 | ,878 |

**Test of Homogeneity**

| | Chi-square (Q statistic) | df | Sig. |
| --- | --- | --- | --- |
| Overall | 42,731 | 13 | <,001 |

**Heterogeneity Measures**

| Overall | Tau-squared | ,134 |
| --- | --- | --- |
| | H-squared | 3,180 |
| | I-squared (%) | 68,6 |

**Figure 7.** Outputs.

Figure 7 shows that the mean effect size estimate was 0.644 (95% CI:.411,.878) and statistically significant ($p < 0.001$). The estimated Hedges' *g* value (0.644) corresponds to a medium-level positive effect according to Cohen [51]. For heterogeneity, Q-statistics, Tau-squared, H-squared, and I-squared values should be examined. The Q-statistics (Q = 42.731, df = 13, $p < 0.001$) was found to be statistically significant. In addition, Tau-squared, H-squared, and I-squared values were found to be 0.134, 3.18, and 68.6, respectively. As a result, there is a statistically significant heterogeneity between studies. Another way of checking the heterogeneity is to create a forest plot (see Figure 8). As shown in Figure 8, individual studies appeared to be distributed heterogeneously. In this case, researchers may want to conduct the moderator analysis that will be shown in Example 3.

*2.2. Example 2 (Odds Ratio)*

In the previous example, it was explained how to conduct the meta-analysis with continuous variables. In this section, how to conduct a meta-analysis using an odds ratio or risk ratio is demonstrated. There are treatment and control groups (as in the previous one) in meta-analyses based on odds ratio or risk ratio, but the data is binary. In this type of meta-analysis, studies that report numbers showing whether an event has occurred or not within two groups are included. For this purpose, we used the sample data retrieved from Cummings and Del Beccaro's [52] study. This data set is presented in Table 3.

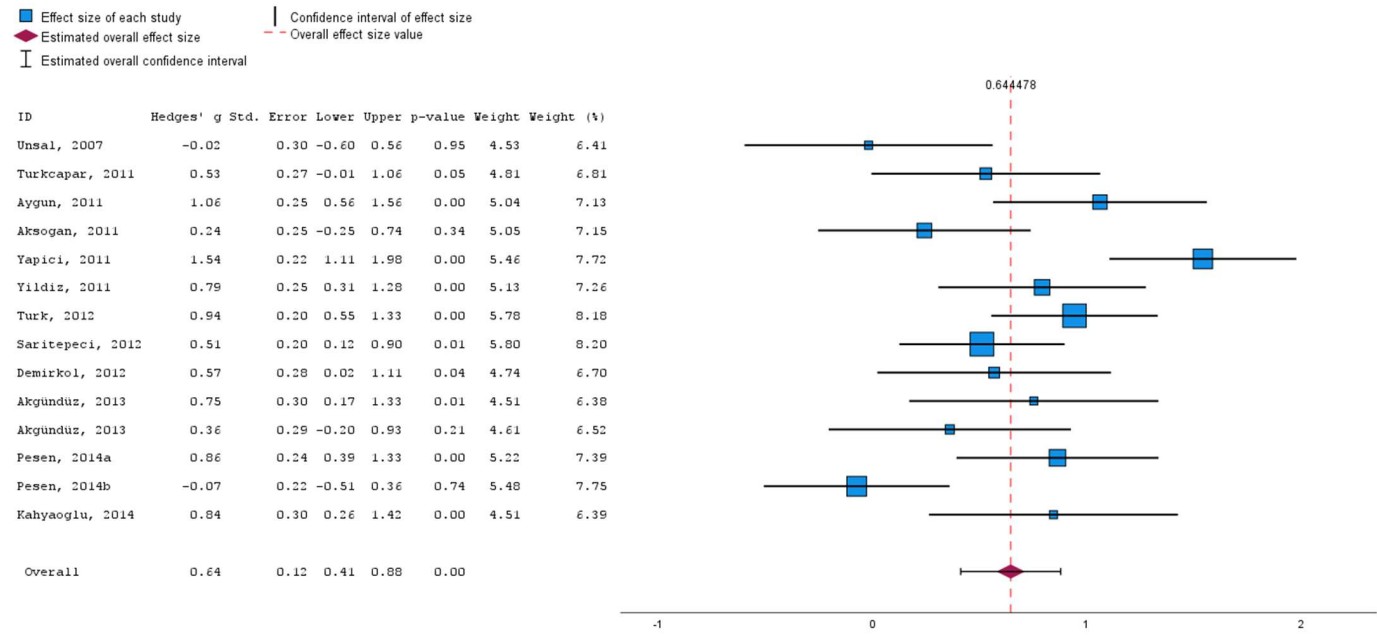

**Figure 8.** Forest Plot.

**Table 3.** Sample data set for binary meta-analysis [52].

| Publication | Antibiotic (Infected) | Antibiotic (Uninfected) | Antibiotic (Total) | Control (Infected) | Control (Uninfected) | Control (Total) |
|---|---|---|---|---|---|---|
| Beelsey, 1975 | 1 | 63 | 64 | 1 | 64 | 65 |
| Day, 1975 | 12 | 44 | 56 | 4 | 52 | 56 |
| Roberts, 1977 | 18 | 187 | 205 | 12 | 88 | 100 |
| Hutton, 1978 | 10 | 132 | 142 | 9 | 134 | 143 |
| Worlock, 1980 | 5 | 66 | 71 | 2 | 32 | 34 |
| Grossman, 1981 | 2 | 172 | 174 | 1 | 90 | 91 |
| Thirlby, 1983 | 16 | 211 | 227 | 17 | 255 | 272 |

The sample data set contains the infected and total numbers of treatment and control group subjects. In seven studies included in the meta-analysis, it was examined whether a simple wound could cause infection with or without antibiotics. Using the data of Roberts and Teddy [53], the odds ratio and log (odds ratio) calculation can be conducted based on the values presented in Table 4.

**Table 4.** Sample Data for Calculation Odds Ratio [53].

| | Infected | Uninfected | Total |
|---|---|---|---|
| **Treatment group** | 18 | 187 | 205 |
| **Control group** | 12 | 88 | 100 |
| **Total** | 30 | 275 | 305 |

Using the information presented in Table 4, the odds ratio and its logarithm (log odds ratio) are calculated as follows:

$$\text{Odds ratio} = \frac{18 \times 88}{187 \times 12} \cong 0.706$$

$$\log_{\text{odss ratio}} = \ln(0.706) \cong -0.348$$

The variance of the odds ratio index is calculated as follows:

$$V_{\text{odds ratio}} = \frac{1}{18} + \frac{1}{187} + \frac{1}{12} + \frac{1}{88} \cong 0.007$$

The odds ratio was calculated as 0.706, indicating that the infection rate among those who use antibiotics is lower than those who do not use antibiotics. The critical odds ratio value is 1.00. Since the odds ratio value is less than 1, the interpretation was made like this.

The risk ratio, log (risk ratio) values, and its variance can be calculated as below.

$$\text{Risk ratio} = \frac{18/205}{12/100} \cong 0.732$$

$$\log_{\text{risk ratio}} = \ln(0.732) \cong -0.312$$

$$V_{\text{risk ratio}} = \frac{1}{18} - \frac{1}{205} + \frac{1}{12} - \frac{1}{100} \cong 0.124$$

In addition, meta-analysis can also be conducted with the risk ratio value, but, in this example, we will use the odds ratio. Like in the previous example, there are two options in SPSS: raw data or pre-calculated effect sizes. In the case of pre-calculated effect size values, one can follow these steps:

- Select Analyze > Meta Analysis > Binary Outcomes > Pre-Calculated Effect Size
- An easier way to do this is to conduct the analysis using raw data when you have the data ready as entered in Excel. For this option, the following steps can be used.
- Select File > Open > Data
- Find the data > Open
- Select Analyze > Meta Analysis > Binary Outcomes > Raw Data
- Add the variables of the experimental group (success and failure) into the 'Treatment Group' box
- Add the variables of the control group (success and failure) into the 'Control Group' box
- Add the identifying variable (Authors' names [Study]) into the 'Study ID' box
- Select the effect size type as 'Log Odds Ratio' in the 'Effect Size' box
- Select the model type 'Random-effects' under the 'Model' box
- Open 'Print' Dialogue > Select the 'Test of homogeneity' and 'Heterogeneity Measures' > Click 'Continue' (see Figure 9)

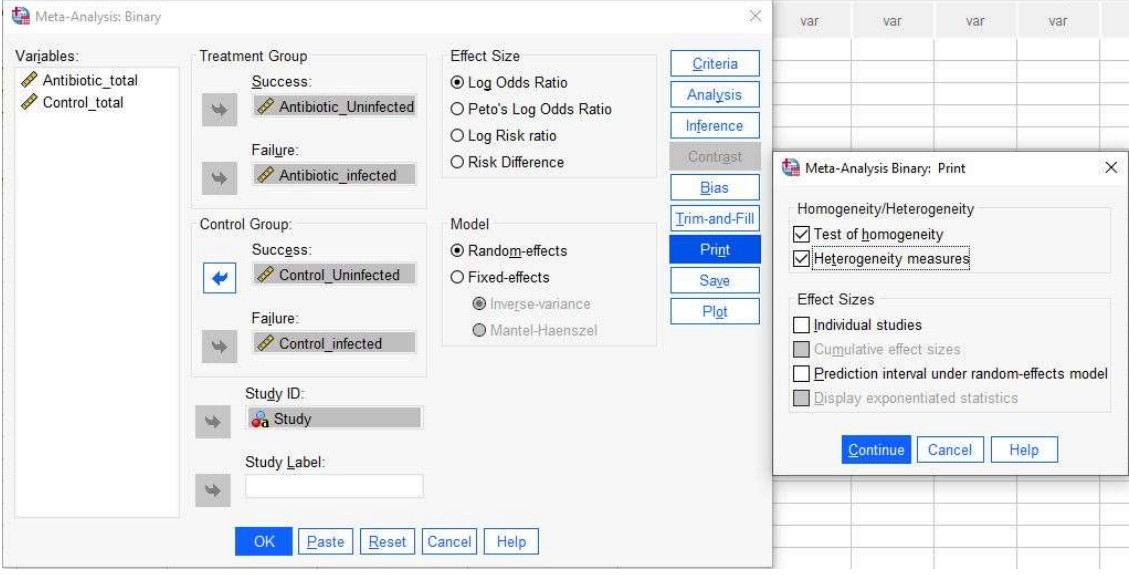

**Figure 9.** Data identification menu.

- Open 'Plot' Dialogue > Select 'Forest Plot' box and all 'Display Columns' boxes > Select 'Overall effect size' box > Click 'Continue'
- Click 'OK' (see Figure 10)

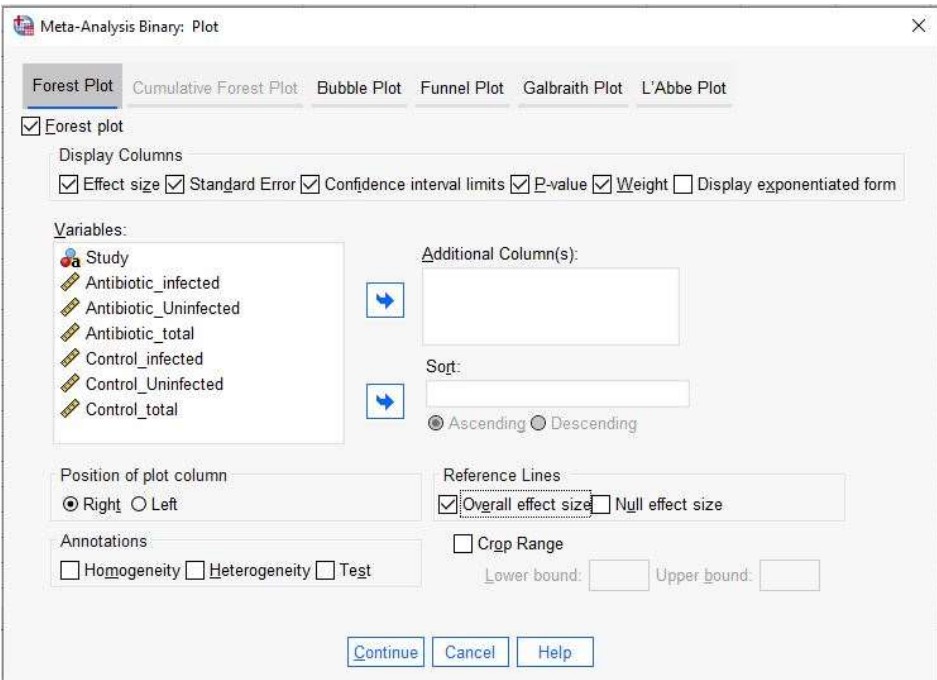

**Figure 10.** Forest Plot Menu.

When these steps are applied, the outputs will be presented in the new window as in Figure 11.

**Meta-Analysis Summary**

| Data Type | Raw |
|---|---|
| Outcome Type | Binary |
| Effect Size Measure | Log Odds Ratio |
| Model | Random-effects |
| Weight | Inverse-variance[a] |
| Estimation Method | REML |
| Standard Error Adjustment | None |

a. Random-effects weights including both within- and between-study variance.

**Effect Size Estimates**

| | Effect Size | Std. Error | Z | Sig. (2-tailed) | 95% Confidence Interval Lower | Upper |
|---|---|---|---|---|---|---|
| Overall | -,127 | ,2077 | -,609 | ,542 | -,534 | ,281 |

**Test of Homogeneity**

| | Chi-square (Q statistic) | df | Sig. |
|---|---|---|---|
| Overall | 4,923 | 6 | ,554 |

**Heterogeneity Measures**

| Overall | Tau-squared | ,004 |
|---|---|---|
| | H-squared | 1,012 |
| | I-squared (%) | 1,2 |

**Figure 11.** Output for Binary Data.

As seen in Figure 11, the mean effect size estimate was found to be −0.127 (95% CI: −0.534, 0.281) and statistically non-significant (*p* = 0.542). Additionally, for heterogeneity, Q-statistics, Tau-squared, H-squared, and I-squared values should be examined. The Q-statistics (Q = 4.923, df = 6, *p* = 0.554) value was found to be statistically non-significant. In addition, Tau-squared, H-squared, and I-squared values were estimated to be 0.004, 1.012, and 1.2, respectively. The forest plot is presented in Figure 12.

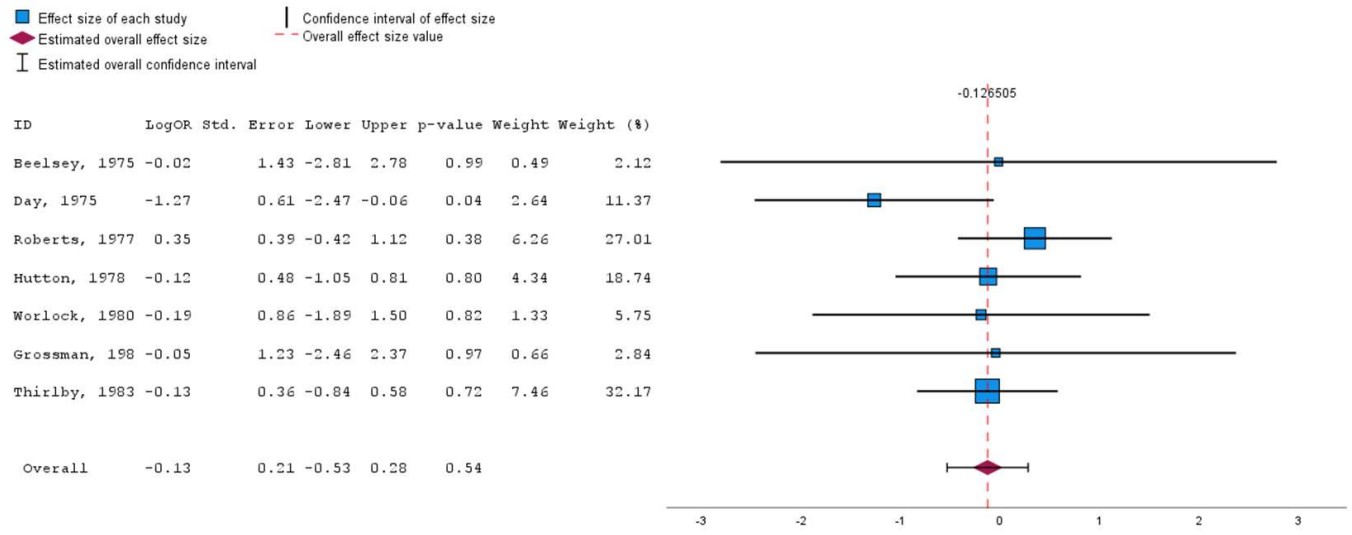

**Figure 12.** Forest Plot for Binary Data.

### 2.3. Example 3 (Correlation)

In this example, we will show you how to conduct a meta-analysis based on correlational data. Correlational meta-analyses are used to find the overall correlation estimate between two continuous variables. For example, researchers may want to examine the relationship between schizotypy and creativity as in Acar and Sen [54]. In this case, a meta-analyst should collect Pearson correlation and sample size values. However, Pearson correlation (*r*) cannot be used directly in meta-analysis due to its dependency on its own variance (see [4]). Thus, Pearson correlation values should be transformed to Fisher's *z* values using the following equation [55]:

$$z = 0.5 \times \ln\left(\frac{1+r}{1-r}\right), \qquad (7)$$

where *r* represents the Pearson correlation value. In addition, the variance of the Fisher's Z-transformed correlations can be calculated as

$$V_z = \frac{1}{n-3} \qquad (8)$$

where *n* represents the sample size. The SPSS program does not have an option to calculate Fisher's Z-transformed correlations and its variance. Thus, the users need to compute these values. A simple-to-use Excel function called FISHER() can be used for this purpose. Another option would be using online calculators (https://www.campbellcollaboration.org/research-resources/effect-size-calculator.html) (accessed on 30 July 2022). The sample data set presented in Table 1 was used for this empirical example. The data set in Table 1 was taken from Göçen and Şen [48] that showed the overall relationship between organizational commitment and spiritual leadership. Only ten studies were drawn from the original study and are presented in Table 1. As you can see, Fisher's Z-transformed correlations (*z*) and their variances (*vz*) are presented in Table 1. There are four moderator variables

(i.e., country, region, sector, and female percent) in addition to sample size (*n*), Pearson correlation (*r*), and its variance (*vr*) (see Figure 13). As always, a variable of study ID was presented in the first column. The screenshot of this data set in SPSS is demonstrated in Figure 13.

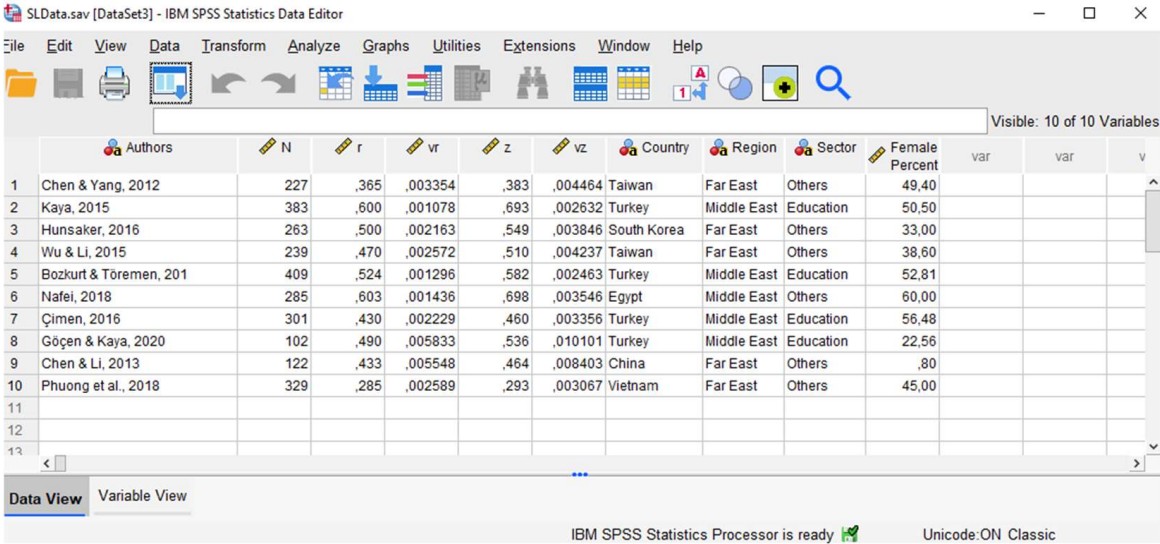

**Figure 13.** Dataset.

To perform the necessary analyses, one has to open the data in SPSS and click on the Meta Analysis menu. For this purpose, the following steps should be performed in the SPSS menu:

- Select Analyze > Meta Analysis > Continuous Outcomes > Pre-Calculated Effect Size
- Add the effect size variable (e.g., *z*) into the 'Effect Size' box
- Add the variance variable (e.g., *vz*) into the 'Variance' box
- Add the identifying variable (e.g., authors) into the 'Study ID' box
- Select the model type as random-effects under the 'Model' box
- Click 'OK' (see Figure 14)

When these steps are applied, the mean effect size estimate can be obtained as 0.519 (95% CI: 0.436, 0.602). This estimate was found to be statistically significant ($p < 0.001$). In order to interpret this value, one needs to retransform this mean value into Pearson correlation. This can be achieved with the following formula:

$$r = \frac{e^{2z} - 1}{e^{2z} + 1}. \tag{9}$$

Alternatively, an easy-to-use Excel function called FISHERINV() can be used for this purpose. As applied in Excel, FISHERINV(0.519) yields the mean effect size as 0.477 in terms of Pearson correlation. To perform the heterogeneity analyses, one has to click on the 'Print' dialog on the main screen shown in Figure 14. The boxes called 'test of homogeneity' and 'heterogeneity statistics' should be checked. When the necessary analyses were performed in SPSS, the Q-statistics value was found to be 44.468 (df = 9, $p < 0.001$). In addition, Tau-squared, H-squared, and I-squared values were found to be 0.014, 4.534, and 77.9, respectively. As a result, there is a significant and a large amount of heterogeneity between studies. As stated above, the 'Plot' dialog can be used to obtain several plots, including forest plots and funnel plots. The forest plot presented in Figure 15 also shows the heterogeneity between studies.

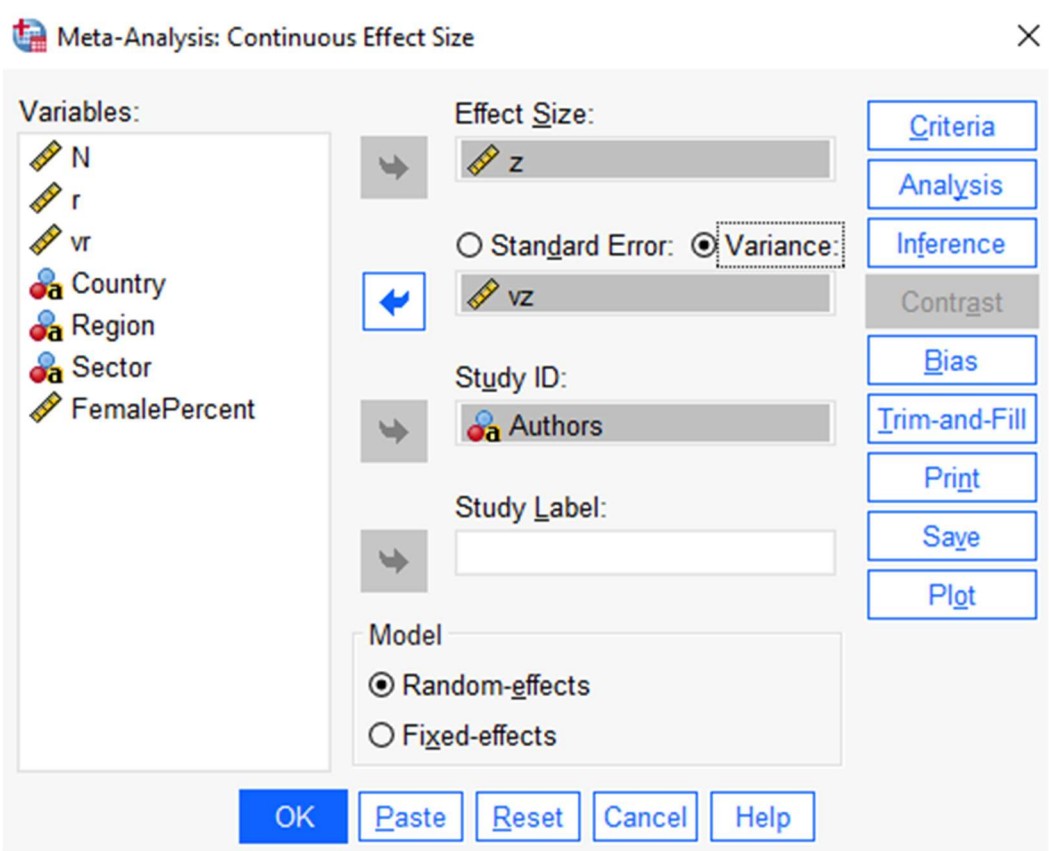

**Figure 14.** Data Identification Menu.

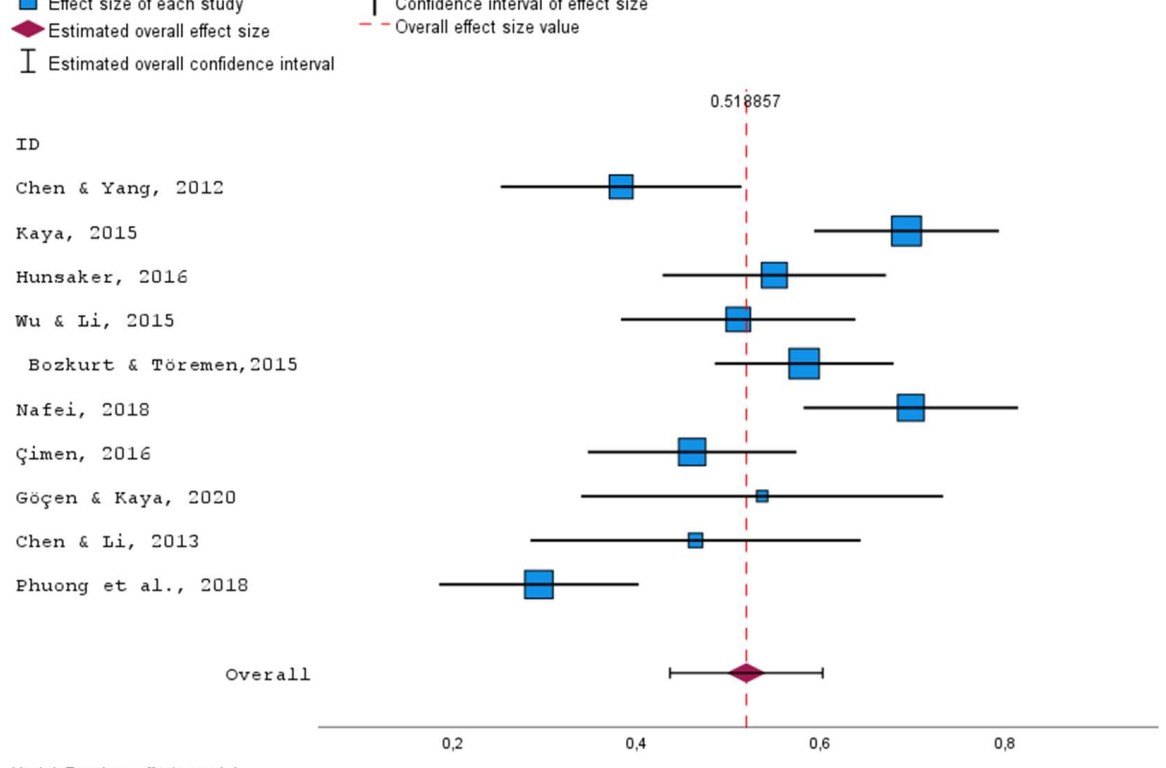

**Figure 15.** Forest Plot for Correlation Data.

Publication bias was also assessed using the funnel plot presented in Figure 16. To perform Egger's test, the 'Bias' dialog was used, and the intercept value was estimated as 0.607 ($p$ = 0.014). When the 'Trim-and-fill' dialog was selected, the results suggested that no imputation was needed based on the slopes of Egger's test. Thus, publication bias was not a concern for the example data set.

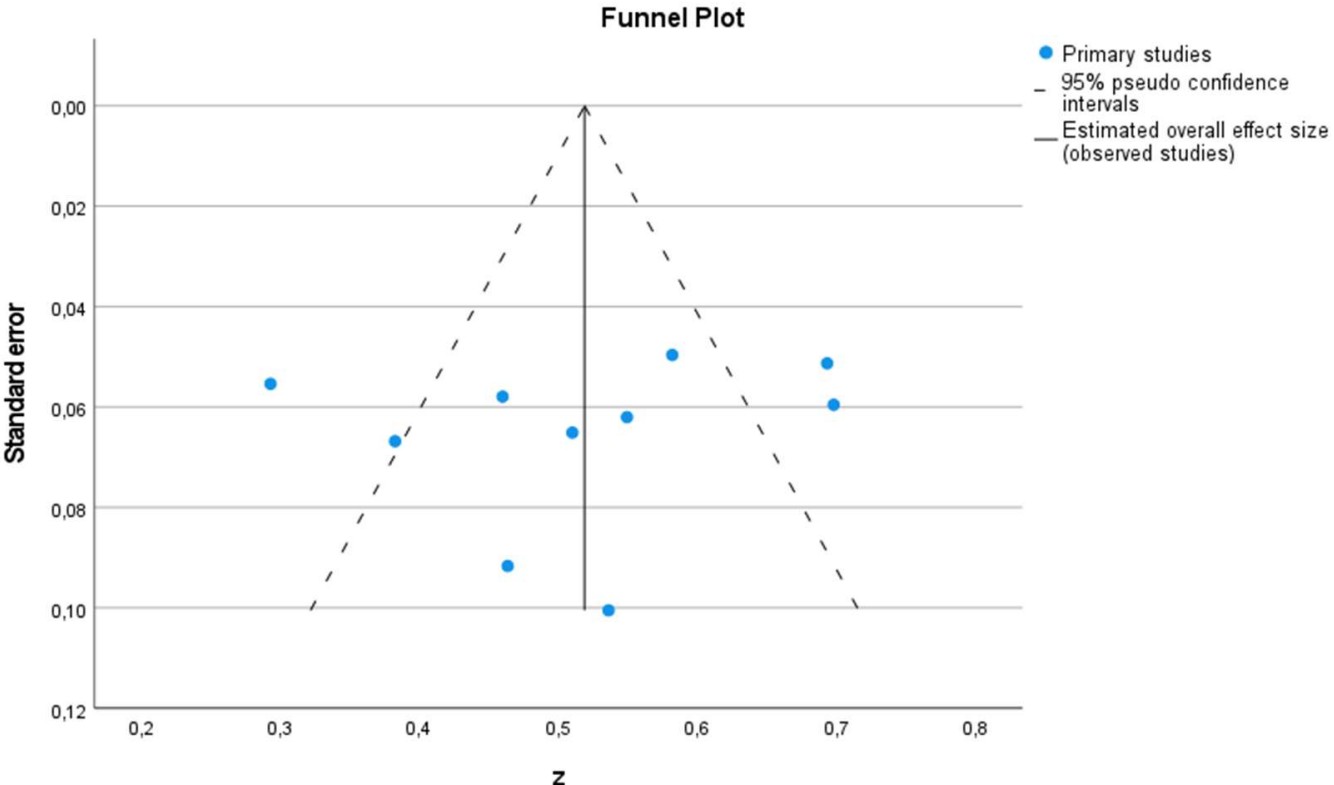

**Figure 16.** Funnel Plot for Correlation Data.

Given the statistically significant heterogeneity, it would be a good idea to conduct subgroup analysis and moderator analysis. The final part of this example shows the application of moderator analyses with categorical (region) and continuous (female proportion) variables listed in Table 1. To perform the subgroup analysis, one needs to open the 'Analysis' dialog when the screen presented in Figure 14 is open. Under the 'Analysis' dialog, you need to add the categorical variable (e.g., region) into the 'Subgroup Analysis' box (see Figure 17).

The results of subgroup analysis with region variable are presented in Figure 18. As shown in Figure 18, there is a statistically significant difference between the mean effect size values of studies conducted in the Far East and the Middle East ($Q_{(1)}$ = 5.564, $p$ = 0.018). The average effect size for studies that were conducted in the Middle East (.599) was significantly higher than the average effect size for studies that were conducted in the Far East (0.436). Additionally, the variance within the Far East studies indicated statistically significant heterogeneity ($Q_W$ = 11.957, $df$ = 4, $p$ = 0.018), similar to the variance within Middle East studies ($Q_W$ = 12.346, $df$ = 4, $p$ = 0.015).

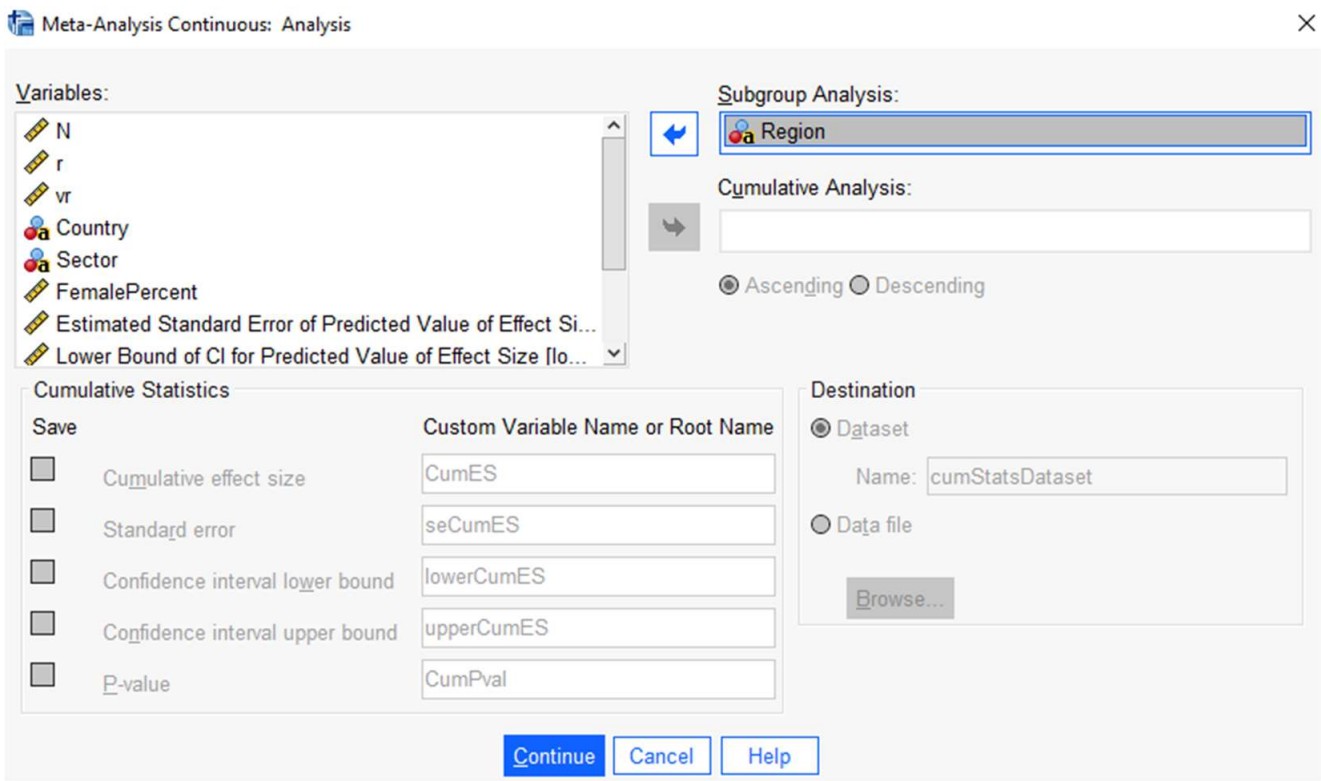

**Figure 17.** Subgroup Analysis Screen.

## Effect Size Estimates for Subgroup Analysis

| | Effect Size | Std. Error | Z | Sig. (2-tailed) | 95% Confidence Interval Lower | 95% Confidence Interval Upper | 95% Prediction Interval[a] Lower | 95% Prediction Interval[a] Upper |
|---|---|---|---|---|---|---|---|---|
| Far East | ,436 | ,0501 | 8,711 | ,000 | ,338 | ,534 | ,110 | ,762 |
| Middle East | ,599 | ,0476 | 12,579 | ,000 | ,506 | ,693 | ,286 | ,913 |
| Overall | ,519 | ,0424 | 12,239 | ,000 | ,436 | ,602 | ,232 | ,805 |

a. Based on t-distribution.

## Test of Homogeneity

| | Chi-square (Q statistic) | df | Sig. |
|---|---|---|---|
| Far East | 11,957 | 4 | ,018 |
| Middle East | 12,346 | 4 | ,015 |
| Overall | 44,468 | 9 | <,001 |

## Test of Subgroup Homogeneity

| | Chi-square (Q statistic) | df | Sig. |
|---|---|---|---|
| Region | 5,564 | 1 | ,018 |

**Figure 18.** Results of Subgroup Analysis with Region Variable.

To perform the meta-regression analysis with both categorical and continuous variables, the following steps should be performed in the SPSS menu:

- Select Analyze > Meta Analysis > Meta Regression
- Add the effect size variable (e.g., *z*) into the 'Effect Size' box
- Add the variance variable (e.g., *vz*) into the 'Variance' box
- Add the continuous variable (e.g., FemalePercent) into the 'Covariate(s)' box
- Add the categorical variable (e.g., Region) into the 'Factor(s)' box
- Select the model type as random-effects under the 'Model' box
- Click 'OK' (see Figure 19)

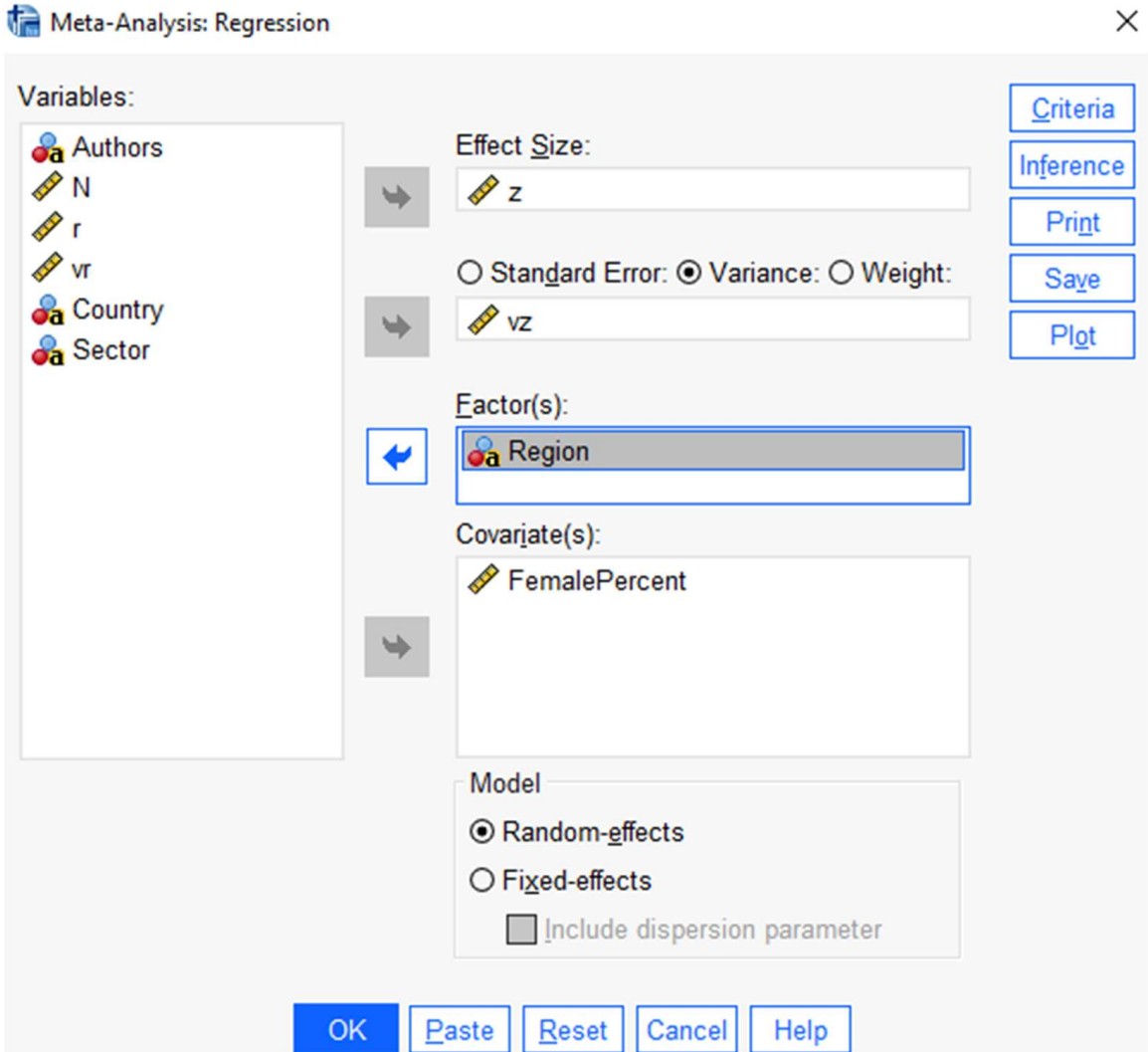

**Figure 19.** Data Identification Menu for Meta Regression.

The results of the meta-regression analysis with female percentage are presented in Figure 20. Meta-regression analyses were performed with a random-effects model using unrestricted maximum likelihood estimation. As shown in Figure 20, results of meta-regression analysis suggested that none of these variables (region and female percentage) were statistically significant predictors of the relationship between organizational commitment and spiritual leadership ($p > 0.05$).

➡️ **Meta-Analysis: Regression**

### Model Summary

| | |
|---|---|
| Effect Size | z |
| Variance | vz |
| Model | Random-effects meta-regression |
| Method | REML |
| SE adjustment | None |

### Case Processing Summary

| | N | Percent |
|---|---|---|
| Included | 10 | 100,0% |
| Excluded | 0 | 0,0% |
| Total | 10 | 100,0% |

### Model Coefficient Test

| Wald Chi-square | df | Sig. |
|---|---|---|
| 5,271 | 2 | ,072 |

Tests the null hypothesis that all coefficients other than the intercept are equal to zero.

### Parameter Estimates

| Parameter | Estimate | Std. Error | t | Sig. (2-tailed) | 95% Confidence Interval Lower | 95% Confidence Interval Upper |
|---|---|---|---|---|---|---|
| (Intercept) | ,646 | ,1407 | 4,589 | ,003 | ,313 | ,978 |
| Region = Far East | -,177 | ,0820 | -2,155 | ,068 | -,371 | ,017 |
| Region = Middle East | 0[a] | . | . | . | . | . |
| FemalePercent | -,001 | ,0026 | -,355 | ,733 | -,007 | ,005 |

a. This parameter is set to zero because it is redundant.

### Test of Residual Homogeneity

| Chi-square (Q statistic) | df | Sig. |
|---|---|---|
| 23,317 | 7 | ,002 |

Tests the null hypothesis that tau-squared is equal to 0.

### Residual Heterogeneity

| | |
|---|---|
| Tau-squared | ,009 |
| I-squared (%) | 69,9 |
| H-squared | 3,320 |
| R-squared (%) | 37,1 |

**Figure 20.** Results of Meta-regression Analysis with Female Percentage.

## 3. Conclusions

The present article is meant to provide a general overview of the capabilities of the IBM SPSS software package for conducting meta-analysis. Therefore, this tutorial article introduced readers to the key features of IBM SPSS Statistics. The steps of meta-analysis using IBM SPSS were described and demonstrated over three examples. In summary, this tutorial covered the following technical considerations necessary for the meta-analysis application in IBM SPSS: creating data sets includes measures of effect sizes and their variances as well as study identifiers, choosing appropriate options, estimating the mean effect sizes (Hedges' *g*, odds ratio, and correlation), checking the heterogeneity, creating the plots, assessment of publication bias, and conducting moderator analyses via subgroup analysis and meta-regression model.

As it is known, several software packages are used for meta-analysis. Among these software packages, there are those that are used only for meta-analysis, those that work as a submenu of comprehensive software, macros, and statistical packages, paid or free ones. IBM SPSS is a comprehensive but paid statistical program that offers a 30-day trial version. While some of the statistical analyses for meta-analysis were possible with SPSS Macros until the latest version (see [47]), a meta-analysis submenu was added into SPSS28. It would be useful to compare IBM SPSS with other meta-analysis software packages to better understand its features. A comparison of the capabilities of the IBM SPSS, CMA, and metafor packages for conducting meta-analyses is presented in Table 5 as in [56]. As shown in Table 5, IBM SPSS can be considered in between the CMA and R metafor package in terms of the meta-analysis capabilities. There are several options for meta-analysis applications. In addition to other properties not listed in Table 5, IBM SPSS has most of the features listed in Table 5. For example, IBM SPSS Statistics has options for Glass' delta, which is not available in most of the other packages. Another positive aspect is that it allows analysis by entering both raw data and pre-calculated effect size. However, its current version does not have options for likelihood ratio tests and permutation tests as in the metafor package. Another limitation of IBM SPSS is that it does not allow simultaneous analysis of different data formats as in CMA software. Perhaps one of the most important shortcomings in SPSS28 is the ability to calculate the effect size for only one measurement (e.g., posttest) of the two groups in the standardized mean difference. In this case, it is necessary to calculate the effect size with online calculation tools and enter pre-calculated effect sizes into the SPSS28. Despite these limitations, it is clear that IBM SPSS will be among the main programs to be preferred by meta-analysis practitioners for future research in psychology and other areas. This is mainly because it is relatively straightforward and user-friendly, so this tutorial is intended to be a basic guide for first-time users who wish to familiarize themselves with the meta-analysis capabilities of IBM SPSS.

**Table 5.** Comparison of the Capabilities of the IBM SPSS, CMA, and Metafor Packages for Conducting Meta-analyses.

|  | IBM SPSS | CMA | Metafor |
|---|---|---|---|
| **Model fitting:** | | | |
| Fixed-effect models | yes | yes | yes |
| Random-effects models | yes | yes | yes |
| Heterogeneity estimator | various | various | various |
| Mantel–Hanszel method | yes | yes | yes |
| Peto's method | yes | yes | yes |
| **Plotting:** | | | |
| Forest plots | yes | yes | yes |
| Funnel plots | yes | yes | yes |
| Radial plots | yes | no | yes |
| L'Abbe plots | yes | no | no |
| Q-Q normal plots | yes | no | yes |

**Table 5.** *Cont.*

|  | IBM SPSS | CMA | Metafor |
|---|---|---|---|
| **Moderator analyses:** | | | |
| Categorical moderators | single * | single | multiple |
| Continuous moderators | multiple * | multiple | multiple |
| Mixed-effects models | yes | yes | yes |
| **Testing/Confidence Intervals:** | | | |
| Knapp and Hartung adjustment | yes | yes | yes |
| Likelihood ratio tests | no | no | yes |
| Permutation tests | no | no | yes |
| **Other:** | | | |
| Leave-one-out analysis | no | yes | yes |
| Influence diagnostics | yes | yes | yes |
| Cumulative meta-analysis | yes | yes | yes |
| Tests for funnel plot asymmetry | yes | yes | yes |
| Trim-and-fill method | yes | yes | yes |
| Selection models | no | no | no |
| Prediction interval | yes | no | yes |

* The number of moderators that can be analyzed simultaneously.

We hope that this presentation, along with the screenshots and available data presented in tables, helps psychological researchers to learn and appropriately apply meta-analyses in IBM SPSS. We also hope this tutorial article fosters increased awareness, knowledge, and skills in relation to meta-analysis and sparks further enthusiasm for adding meta-analysis to the methodological toolbox in psychology and other areas.

**Supplementary Materials:** The following supporting information can be downloaded at: https://www.mdpi.com/article/10.3390/psych4040049/s1.

**Author Contributions:** S.S.: Conceptualization, Methodology, Writing, and Formal Analysis. I.Y.: Methodology, Formal Analysis, Writing—Review and Editing. All authors have read and agreed to the published version of the manuscript.

**Funding:** This research received no external funding.

**Institutional Review Board Statement:** Not applicable.

**Informed Consent Statement:** Not applicable.

**Data Availability Statement:** Data is contained within the article.

**Conflicts of Interest:** The authors declare no conflict of interest.

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
