# Peer review of "A Tutorial on How to Conduct Meta-Analysis with IBM SPSS Statistics"

_psych, doi:10.3390/psych4040049_

Round 1
Reviewer 1 Report
The manuscript demonstrated the step-by-step hands-on of conducting a meta-analysis in SPSS. As a researcher who uses meta-analysis in my work, and also as a teacher who coaches fellow students to work on their meta-analysis, I found this work very useful and informative. I have a few minor comments for the authors to consider.
Introduction:
- I have mixed feelings towards the statement “While this method is called meta-synthesis within the scope of qualitative research, it can be applied as a meta-analysis in the context of combining the results of quantitative research” (p. 2, lines 54-56). To some scholars, a systematic review is not entirely equivalent to meta-analysis. It has been suggested that systematic review refers to a set of procedures for conducting a literature review, which may or may not include a quantitative synthesis (i.e. meta-analysis) (ref Chapter 10 of the Cochrane Handbook for Systematic Reviews of Interventions). Clarification of this statement is recommended.
- I suggest putting the statement “Any standardized index (standardized mean difference, correlation, and odds ratio) can be used as an effect size as long as it is comparable between studies, independent of the sample, and indicates the size and direction of the effect” (p. 2, lines 71-74) right after the statement “In order to conduct a meta-analysis study, Glass suggested using the effect size value when combining the findings of multiple studies correctly [13,14]” (p.2, 67-68), so that a definition of “effect sizes” will come naturally after when the term first appears in line 68.
- There is a module called “MAJOR” in JAMOVI, a free statistical software becoming more popular nowadays. As far as I know, the module is developed from the R package metafor. This can be added as an additional example to the readers in the first paragraph on p. 4.
- Section 1.3 is a bit hard to follow. Maybe adding a few subtitles (e.g. main analysis, publication bias analysis and subgroup analysis & meta-regression) would help organize this section better.
Empirical examples:
- I wonder if the raw data for the three examples could be provided as sav files and prepared as supplemental materials, or uploaded to OSF? This could save space for tables (Table 2 & 3) and the figure (Figure 13) in the main text, without a loss of these useful materials for the demonstration.
Conclusion:
- I had previous experiences with Comprehensive Meta-analysis (CMA) and remembered that L’Abbe plots are not available in CMA (ref Table 5). But maybe I am wrong.
- In Table 5, I am confused about “single/multiple” for the rows “categorical moderators” and “continuous moderators”. A note should be put underneath the table to clarify the meaning of these terms.
- It has been increasingly popular to report the prediction interval of the pooled effect sizes in a meta-analytic review. Therefore, it would be informative to add the information about which software/ packages can compute automatically the prediction intervals for their users in Table 5.
Author Response
The manuscript demonstrated the step-by-step hands-on of conducting a meta-analysis in SPSS. As a researcher who uses meta-analysis in my work, and also as a teacher who coaches fellow students to work on their meta-analysis, I found this work very useful and informative. I have a few minor comments for the authors to consider.
Authors’ response: Thank you so much for your constructive critics and annotations on our study. We worked through your comments. Please see our explanations below.
Introduction:
- I have mixed feelings towards the statement “While this method is called meta-synthesis within the scope of qualitative research, it can be applied as a meta-analysis in the context of combining the results of quantitative research” (p. 2, lines 54-56). To some scholars, a systematic review is not entirely equivalent to meta-analysis. It has been suggested that systematic review refers to a set of procedures for conducting a literature review, which may or may not include a quantitative synthesis (i.e. meta-analysis) (ref Chapter 10 of the Cochrane Handbook for Systematic Reviews of Interventions). Clarification of this statement is recommended.
Authors’ response: Thank you for your insight. Aforementioned sentence was removed. The following sentence was added instead:
However, meta-analysis differs from systematic review in that it only focuses on quantitative studies.
- I suggest putting the statement “Any standardized index (standardized mean difference, correlation, and odds ratio) can be used as an effect size as long as it is comparable between studies, independent of the sample, and indicates the size and direction of the effect” (p. 2, lines 71-74) right after the statement “In order to conduct a meta-analysis study, Glass suggested using the effect size value when combining the findings of multiple studies correctly [13,14]” (p.2, 67-68), so that a definition of “effect sizes” will come naturally after when the term first appears in line 68.
Authors’ response: Revised as suggested.
- There is a module called “MAJOR” in JAMOVI, a free statistical software becoming more popular nowadays. As far as I know, the module is developed from the R package metafor. This can be added as an additional example to the readers in the first paragraph on p. 4.
Authors’ response: Thank you for your suggestion. The following sentences were added to revised manuscript:
A module called MAJOR in Jamovi developed by Kyle Hamilton allows users to do a meta-analysis in using different types of input (e.g., effect sizes, correlations coefficients). Similarly, another open-source statistical software called JASP can also be used for meta-analysis. The engine behind these two software packages comes from the R package metafor.
- Section 1.3 is a bit hard to follow. Maybe adding a few subtitles (e.g. main analysis, publication bias analysis and subgroup analysis & meta-regression) would help organize this section better.
Authors’ response: Thank you for your suggestion. This section was revised as suggested (please see page 6).
Empirical examples:
- I wonder if the raw data for the three examples could be provided as sav files and prepared as supplemental materials, or uploaded to OSF? This could save space for tables (Table 2 & 3) and the figure (Figure 13) in the main text, without a loss of these useful materials for the demonstration.
Authors’ response: Thank you for your suggestion. We prefer to keep those tables in the manuscript for better understanding of the data structure of the meta-analysis. As this is a tutorial paper, it would be better for readers to see them as a part of the manuscript rather than separate files provided in a repository.
Conclusion:
- I had previous experiences with Comprehensive Meta-analysis (CMA) and remembered that L’Abbe plots are not available in CMA (ref Table 5). But maybe I am wrong.
Authors’ response: Yes, you are right. Thanks for your correction. Table 5 was revised accordingly.
- In Table 5, I am confused about “single/multiple” for the rows “categorical moderators” and “continuous moderators”. A note should be put underneath the table to clarify the meaning of these terms.
Authors’ response: We meant number of moderators (single moderator and multiple moderators) by single/multiple. A short note was added underneath the table to clarify the meaning of these terms.
- It has been increasingly popular to report the prediction interval of the pooled effect sizes in a meta-analytic review. Therefore, it would be informative to add the information about which software/ packages can compute automatically the prediction intervals for their users in Table 5.
Authors’ response: Thank you for your suggestion. Another line about prediction intervals was added to end of Table 5.
Reviewer 2 Report
The manuscript provides tutorial using IBM SPSS version 28 for Meta-Analysis. Therefore the article along with SPSS manual could be used for for analysis of data from published articles
It would be better if author could discuss more about the use fixed and random effects, and publication bias.
Author Response
The manuscript provides tutorial using IBM SPSS version 28 for Meta-Analysis. Therefore the article along with SPSS manual could be used for analysis of data from published articles
It would be better if author could discuss more about the use fixed and random effects, and publication bias.
Authors’ response: Thank you for your suggestion. Two paragraphs about fixed/random effects and publication bias were added to Introduction section.
Reviewer 3 Report
This is a useful step by step guide to the use of SPSS in the conduct of meta analyses. This funtionality has been added to the widely-used package SPSS and many readers will find this guide useful to conduct their own meta analysis.
I found the paper provided a useful overview of the history of this analytical technique, providing some context into the criticisms of its use. There may be potential to add some additional comments regarding the robustness tests available to test and adjust for publication bias and outlier publications.
Overall, a nice paper, well presented and will be read and cited widely.
Author Response
This is a useful step by step guide to the use of SPSS in the conduct of meta analyses. This functionality has been added to the widely-used package SPSS and many readers will find this guide useful to conduct their own meta-analysis.
I found the paper provided a useful overview of the history of this analytical technique, providing some context into the criticisms of its use. There may be potential to add some additional comments regarding the robustness tests available to test and adjust for publication bias and outlier publications.
Overall, a nice paper, well presented and will be read and cited widely.
Authors’ response: Thank you for your suggestion. A paragraph about publication bias was added to Introduction section.